# Effect of Material and Process Variables on Characteristics of Nitridation-Induced Self-Formed Aluminum Matrix Composites—Part 1: Effect of Reinforcement Volume Fraction, Size, and Processing Temperatures

**DOI:** 10.3390/ma13061309

**Published:** 2020-03-13

**Authors:** Dae-Young Kim, Pil-Ryung Cha, Ho-Seok Nam, Hyun-Joo Choi, Kon-Bae Lee

**Affiliations:** School of Advanced Materials Engineering, Kookmin University, Seoul 02707, Korea; kdy6603@kookmin.ac.kr (D.-Y.K.); cprdream@kookmin.ac.kr (P.-R.C.); hsnam@kookmin.ac.kr (H.-S.N.)

**Keywords:** aluminum matrix composites, aluminum nitride, nitridation, exothermic, NISFAC process

## Abstract

This paper investigates the effect of the size and volume fraction of SiC, along with that of the processing temperature, upon the nitridation behavior of aluminum powder during the nitridation-induced self-formed aluminum composite (NISFAC) process. In this new composite manufacturing process, aluminum powder and ceramic reinforcement mixtures are heated in nitrogen gas, thus allowing the exothermic nitridation reaction to partially melt the aluminum powder in order to assist the composite densification and improve the wetting between the aluminum and the ceramic. The formation of a sufficient amount of molten aluminum is key to producing sound, pore-free aluminum matrix composites (AMCs); hence, the degree of nitridation is a key factor. It was demonstrated that the degree of nitridation increases with decreasing SiC particle size and increasing SiC volume fraction, thus suggesting that the SiC surface may act as an effective pathway for nitrogen gas diffusion. Furthermore, it was found that effective nitridation occurs only at an optimal processing temperature. When the degree of nitridation is insufficient, molten Al is unable to fill the voids in the powder bed, leading to the formation of low-quality composites with high porosities. However, excessive nitridation is found to rapidly consume the nitrogen gas, leading to a rapid drop in the pressure in the crucible and exposing the remaining aluminum powder in the upper part of the powder bed. The nitridation behavior is not affected by these variables acting independently; therefore, a systematic study is needed in order to examine the concerted effect of these variables so as to determine the optimal conditions to produce AMCs with desirable properties for target applications.

## 1. Introduction

Metal matrix composites (MMCs) containing variously shaped ceramic reinforcements (e.g., particulates, whiskers, and fibers) exhibit superior properties to those of each individual component, simultaneously displaying the good ductility and toughness of the metallic matrix and the high strength and stiffness of the ceramic reinforcements [1,2,3]. Moreover, the significant differences in terms of physical, thermal, electrical, and mechanical properties between the metallic matrix and ceramic reinforcements facilitate the design of tailored MMCs encompassing a wide range of metallic and ceramic properties by combining various choices of matrix and reinforcement materials and varying the size, shape, and volume fraction of the ceramic reinforcements [1,2,3,4,5,6,7,8,9,10]. Recently, considerable technological advances in the manufacture of MMCs have enabled the use of these composites not only in high-tech industries such as aerospace, recreation, and infrastructures but also in everyday life [1,2,10,11,12,13,14,15]. 

Among several commercial MMCs, the aluminum matrix composites (AMCs) occupy approximately 70% of the global market. A variety of AMC manufacturing processes have been developed over the decades, with representative commercial processes including stir casting, infiltration, and powder metallurgy [1,12,15]. Generally, the ceramic reinforcement should be embedded and dispersed in the aluminum matrix to produce the AMC. However, poor wettability between the aluminum and the ceramic reinforcement makes this difficult. Efforts have therefore been made to overcome this inherent issue by high-energy stir casting, high-pressure infiltration of molten aluminum into a preform, or consolidation of a mixture of aluminum and reinforcement powders (powder metallurgy) [1,2,3,4,5,6,7,8,9,10,11,12,13,14,15]. However, these commercial methods require complicated multi-step processes along with additional equipment or catalysts, which increase both the processing time and cost of the final products. Furthermore, the nature of the commercial processes places limitations upon the type and volume fraction of the reinforcement that can be used, while each additional process step can have a detrimental effect on the properties of the final product.

Due to continued technological development, the growth rate of the AMC market is projected to be around 6.9% per year up to 2022 [11]. Nevertheless, AMCs still occupy a small portion of the lightweight structured materials market. The most critical barrier to the commercialization of AMCs is the aforementioned high processing costs. In this respect, recent research has focused on the development of low-cost processes for large-scale AMC manufacture. The development of a simple manufacturing process for AMCs may reduce the process cost and extend the application fields for AMCs. 

In a previous study, we developed a new process for AMCs (namely, the nitridation-induced self-formed aluminum composite or NISFAC process) [16,17], which overcomes the issue of poor wettability between the aluminum and the ceramic reinforcement. In this process, composites can be produced for any type, size, and volume fraction of reinforcement simply by heating a mixture of aluminum powder and ceramic reinforcement under a nitrogen atmosphere in a sealed furnace to prevent the ingress of air. During heating in nitrogen gas, the oxide layer on the surface of the aluminum powder is transformed into nitrides, which greatly improves the wettability between the aluminum and the ceramic reinforcement. Furthermore, the exothermic reaction of nitridation can lower the process temperature to below the melting point of aluminum. Since the NISFAC process is the simplest AMC manufacturing route and has no limitations regarding the selection of reinforcement, it is expected to be sufficiently cost-effective and expand the scope of application. Furthermore, this process is totally different from the conventional AMC manufacturing processes and is expected to inspire researchers to investigate a variety of new topics.

Suitable control of the degree of nitridation is important if sound AMCs are to be produced using the NISFAC process. Hence, it is necessary to conduct a systematic study on the effects of various material characteristics (e.g., the composition of the aluminum powder and the type, size, and volume fraction of the reinforcement) and process variables (e.g., processing temperature and time, and amount of injected nitrogen gas) on the degree of nitridation. The effect of these variables upon the nitridation behavior is reported for the first time in the present study. 

## 2. Materials and Methods

The NISFAC process is a very simple and innovative route to fabricating most contemporary AMCs. As shown in Figure 1, it involves the following two or three simple steps: (i) mixing the raw material powders, (ii) heating under a nitrogen atmosphere, and (optionally) (iii) secondary working. For example, this process can be used to produce Al/SiC composites by first mixing aluminum powder and SiC with the target composition. Any kind of mixing (hand mixing, roll mixing, ball mixing, etc.) can be used in this step. Since the dispersion of the reinforcement is determined at the mixing stage, the selection of an appropriate mixing method to establish the conditions for uniform dispersion will enable the production of composites with a highly uniform distribution of the reinforcement. Thus, the degradation of AMC properties due to poor dispersion of the reinforcement will be avoided.

In this study, we produced Al 6061 alloy matrix composites containing SiC particulates. The average particle size of the aluminum powder (Chengdu Best New Materials Co., Ltd, Sichuan, China) was 7.18 μm, and the average size of the SiC particulate (Showa Denko, Toyama, Japan) was 10–40 μm. Raw materials with various volume fractions (0–70 vol.%) of reinforcement were mixed using a Turbula mixer (DM-T2, Daemyoung Enterprise Co. Ltd., Gwangmyeng, Korea). The mixed powder was placed in a graphite crucible, charged into a furnace, heated for 20–60 min at 620–650 °C under a nitrogen atmosphere, and then removed from the furnace and air-cooled. The heating rate was 5 °C/min and the nitrogen flow rate was maintained at 2–6 L/min. The gas was exhausted via a water-filled beaker to inhibit the ingress of external oxygen into the furnace. During heating, nitridation occurred in the mixture bed due to the reaction between the Al powder and the nitrogen atmosphere. The degree of nitridation, which has a critical impact on the characteristics of the AMC, was estimated by measuring the weight of the crucible before and after heating. The temperature change inside the mixed powder bed due to the exothermic nitridation reaction was also measured with the help of data acquisition software (Lutron, SW-U801-WIN) by inserting a thermocouple in the center of the bed. The microstructure and interfacial structure of the synthesized AMCs were examined by optical microscopy (Eclipse LV100ND, Nikon, Minato, Japan) and scanning electron microscopy (SEM; JSM 2001F, JEOL, Akishima, Japan) with energy-dispersive X-ray spectroscopy (EDS). 

## 3. Results and Discussion

### 3.1. Temperature Change of Powder Bed by Nitridation

The composite formation procedure in the NISFAC process is completely different from that of conventional composite manufacturing processes. In the NISFAC process, the AMCs can be formed spontaneously by simply heating a mixture of aluminum powder and reinforcement without involving complicated multi-step processes. Therefore, it is important to understand the composite formation mechanism in the NISFAC process. As shown in Figure 2 and Table 1, we examined the change in temperature of the powder bed and the degree of nitridation when heating mixtures of Al 6061 alloy powder and SiC (20 and 50 vol.% (22.91 and 54.31 wt.%), 10 and 20 μm particle size) under the conditions described in the previous section. Appendix A shows the XRD patterns of the powder beds held at 650 °C for various times. The furnace temperature increased from room temperature (RT) to 650 °C as the furnace was heated for 128 min. The temperature then remained at 650 °C for a further 20–60 min. At the end of that time, the crucible was removed from the furnace and air-cooled to RT. During heating, the increment of temperature for each sample became sluggish as the temperature reached the solidus line of the Al 6061 alloy (582 °C), possibly due to the latent heat of fusion of aluminum powder. The temperature increase subsequently resumed with continued heating up to a powder bed temperature of 646 °C, when all the samples exhibited similar degrees of nitridation (~4.0%). With further heating, however, the powder bed temperature significantly varied according to the size and content of SiC. For the mixture containing 20 vol.% of SiC with a mean particle size of 10 μm, the bed temperature reached a maximum of 656 °C after 40 min of heating, and then decreased to the target temperature of 650 °C after 1 h of heating (black line, Figure 2a,b). However, for the mixture containing 50 vol.% of SiC with the same particle size (red line, Figure 2a), the bed temperature gradually increased from RT to 692 °C during 42 min of heating, rapidly increased to 1173 °C during 2 min of additional heating, and then rapidly decreased again. This difference in the heating profile may relate to the differing degrees of nitridation for the different contents of SiC; the mixture containing 20 vol.% SiC gave a 5.4% degree of nitridation, while the mixture containing 50 vol.% SiC gave a 49.8% degree of nitridation. However, even with a volume fraction of 50%, increasing the SiC particle size to 20 μm delayed the nitriding reaction and reduced the temperature rise compared to that observed for 10 μm SiC. As a result, the degree of nitridation also decreased from 49.8% to 31.3% (Table 1).

In previous work, we reported that the degree of nitridation and resultant powder-bed temperature change varied significantly according to the volume fraction of the reinforcement [16,18,19,20,21]. Before heating, the aluminum powder in the powder bed was covered with an amorphous Al_2_O_3_ layer as shown in Figure 1a. It is well known that the formation of AlN by the reaction between the solid Al_2_O_3_ layer and nitrogen gas according to Equation (1) is thermodynamically impossible below 1000 °C [22,23,24,25,26,27]:
2Al_2_O_3_ (s) + 2N_2_ (g) = 4AlN (s) + 3O_2_ (g),(1)

As found in both the previous [16] and present study, the nitriding reaction begins even at temperatures below the melting point of Al (or the liquidus temperature of aluminum alloys), and the degree of nitridation increases with time (Figure 1 and Table 1), resulting in the transformation of the surface Al_2_O_3_ layers into AlON/AlN layers (Figure 1b). This surface modification of aluminum powder has been previously observed by transmission electron microscopy (TEM) and electron energy loss spectroscopy (EELS) analysis, thus demonstrating that both the surface modification and the temperature rise due to the exothermic reaction could contribute to an enhanced wettability between the aluminum matrix and the reinforcement [16,17,18,19,20,21,28,29,30,31]. 

SEM images of samples produced under the conditions of Figure 2 are presented in Figure 3 and Figure 4 along with photographic images of the samples after lathe working. While all the samples revealed similar degrees of nitridation and bed temperatures by up to 30 min of heating at 650 °C, their densification behaviors differed significantly after cooling. The samples containing 20 vol.% 10 μm SiC particles and 50 vol.% 20 μm SiC particles were sufficiently densified to enable lathe working. However, the upper part of the samples containing 50 vol.% 10 μm SiC particles remained powdery, while the bottom part was densified but brittle. After heating for 1 h, all three types of samples were sufficiently densified to allow machining. However, only the sample with 20 vol.% 10 μm SiC particles exhibited a shiny metallic surface, while the other two samples (those with 50 vol.% 10 μm and 20 μm SiC particles) lacked the metallic surface luster due to the presence of numerous pores arising from excessive nitridation. The sample with 50 vol.% 10 μm SiC particles contained 18.4 g of Al powder (out of 40 g total powder), and the degree of nitridation was 49.8%; hence, only 9.6 g of the Al powder formed the matrix, while the remainder left numerous pores. The evolution of microstructure during heating was also investigated by SEM analysis. As previously reported [16,17,18,19,20,21], the amount of reaction product on the surface of the Al powder and SiC particles increased with increasing heating time, and EDS analysis identified the reaction product as aluminum nitride (AlN) [16]. During the NISFAC process, the heat dissipated by the exothermic nitridation reaction can lead to the melting of the Al powder, and the molten Al can fill the voids in the powder bed to form a dense composite. Hence, in order to produce sound, pore-free composites, a sufficient amount of molten Al is required to fill all the voids. As shown in Table 1 and Figure 2, Figure 3 and Figure 4, this can be ensured by an optimal combination of the degree of nitridation and the melting point of Al powder. Therefore, the effect of processing variables on the porosity of the final composites was investigated.

### 3.2. The Effect of Processing Temperature and Volume Fraction of SiC

In order to examine the effect of processing temperature upon composite fabrication, mixtures of 6061 Al alloy powder (40 g) and 20 μm SiC particles (0–70 vol.%) were heated for 1 h at 620–650 °C in nitrogen at a flow rate of 2 L/min, then air-cooled. Thus, all the processing conditions were fixed except for the processing temperature. Figure 5 shows the crucible images obtained immediately after air-cooling along with images of the composites obtained after machining. While all the composites contain SiC particles of the same size, the images reveal differences according to the processing temperature and volume fraction of SiC. While developing the NISFAC process, we learned empirically that the degree of shrinkage of the powder bed and the visual appearance of the machined composites could be used to estimate the quality (soundness) of the composites produced, and that this could be confirmed by examining optical images of the microstructures after surface polishing. Table 2 shows the degree of nitridation of the composites produced under the conditions in Figure 5. When the degree of nitridation was appropriate for the molten Al to sufficiently fill the voids in the powder bed, the final composites possessed a sufficient amount of metal (Al) without pores, thereby displaying a metallic surface luster after machining. The processing temperature affects the amount of Al to melt and the degree of nitridation. In particular, it can be seen that the effect of the processing temperature on the degree of nitridation is apparent at high volume of 40% or more. The solidus and liquidus of the 6061 Al alloy are 582 °C and 652 °C, respectively [32]. When produced at 650 °C, the degree of nitridation increased rapidly as the volume fraction of SiC increased above 40%, and the composites containing 70 vol.% SiC exhibited 94.2% nitridation, indicating that most of the Al in the powder bed was transformed into AlN. As a result, the final composite showed a dark gray surface instead of a shiny metallic surface and was too brittle to lathe. When the processing temperature was decreased, however, the final composites displayed a glittering metallic luster, and significant shrinkage was observed in the powder bed even for samples containing more than 60 vol.% SiC. In addition, as the processing temperature decreased, the amount of powder remaining un-melted in the upper part of the powder bed increased, so the fabrication conditions were not optimal. Hence, for samples with a different matrix and volume fraction of SiC, the optimal processing temperature should be examined in order to minimize material loss.

The actual temperature inside the powder bed during the NISFAC process is determined by the processing temperature together with an additional contribution from the exothermic nitridation reaction. The processing temperature (620 °C) was higher than the solidus (582 °C) but much lower than the liquidus (652 °C) of 6061 Al alloys. At this temperature, Al did not melt when heated in argon gas but did melt when heated in nitrogen gas, strongly suggesting that the exothermic nitridation reaction provided the powder bed with sufficient additional heat to melt the Al. As shown in Table 2, at 650 °C, the degree of nitridation (corresponding to the increase in temperature due to the exothermic reaction) could be tuned by controlling other processing variables such as the size of Al powder and SiC particles, the heating time, and the nitrogen concentration when the chemical composition of the Al matrix, the type of reinforcement, and the processing temperature are fixed, thus enabling the production of composites with a high volume of SiC particles. Furthermore, since the NISFAC process takes advantage of additional heat generated by the exothermic reaction, the composites can be manufactured at temperatures below the melting temperature of Al (or the liquidus of Al alloys). This reduction in the processing temperature could greatly reduce the energy consumption and processing time, thereby enhancing the economic competitiveness of this method. The low processing temperature may also suppress the formation of unwanted side products, thus enhancing the properties of the composites.

### 3.3. Effect of SiC Particle Size

The effects of the size and volume fraction of SiC particles upon the composites produced at 650 °C are indicated in Figure 6 and Table 3. As explained above, when 20 μm SiC particles were used, excessive nitridation occurred at 50 vol.% or more, while sound composites with metallic surface lusters were produced even for a high volume of SiC when 40 μm SiC particles were employed. Thus, an increase in the SiC particle size suppressed excessive nitridation, leading to sufficient shrinkage of the powder bed and reduction in the quantity of powder remaining in the upper part of the powder bed. However, a further reduction in the SiC particle size to 10 μm (row I, Figure 6) resulted in a deterioration of the quality of the composites. The size of the reinforcement determines the relative porosity of the powder bed. When the size of Al powder and processing temperature were fixed, the porosity and resultant pathway for nitrogen gas decreased with increasing SiC particle size. This suppressed rapid nitridation, thereby leading to the formation of a sufficient amount of molten Al phase. As shown in Table 3, the effect of the size of SiC particles on the degree of nitridation became more obvious when the volume fraction of SiC exceeded 40%. The glowing state of the bed in the crucible differs depending on the degree of nitridation during the process. In addition, some of the upper part of the powder bed displayed a white color when excessive nitridation occurred. The excessive nitridation rapidly consumed the nitrogen gas, leading to a rapid drop in the pressure of the furnace and resultant backflow of cold water (which was intentionally set up to prevent the ingress of air from outside the crucible). Thus, the upper part of the powder bed was quenched by the backflow of cold water, leading to a color change from grey to white.

### 3.4. Microstructures of the Obtained Composites

Several thousand samples were produced under various processing conditions in batches ranging from small quantities (40 g) to large quantities (4 kg). This process is beneficial for large-scale and cost-effective production because it does not require compaction and/or an oxygen-free atmosphere. Nevertheless, the composites manufactured using this new process are expected to exhibit comparable properties to those manufactured following the typical powder metallurgical routes, and better properties than those manufactured following the typical stir casting route. In addition, Figure 7 shows the distribution of SiC particles in the AA 6061 matrix composites for which large quantities of composites were produced, i.e., 500 g (Figure 7a,b) and 1000 g (Figure 7c,d). The images reveal a uniform distribution of the SiC particles regardless of the volume fraction of SiC, thus demonstrating one of the benefits of the NISFAC process, namely that SiC particles can be uniformly mixed with aluminum powder via an appropriate powder mixing technique to provide a homogeneous dispersion in the composite matrix despite the relative simplicity of the technique. Furthermore, since the processing temperature is below the melting point (or liquidus) of the matrix, the formation of unfavorable products at the interface between the reinforcement and the matrix is negligible, potentially leading to enhanced properties compared with those of conventional composites. This aspect will be reported elsewhere.

## 4. Conclusions

The newly developed NISFAC process was used to investigate the effects of the size and volume fraction of SiC particles and processing temperature upon the nitridation behavior and the quality of the obtained composites. Since the NISFAC process is based on the nitridation of aluminum, control of the degree of nitridation is the most important factor. The degree of nitridation increased with decreasing SiC particle size and increasing SiC volume fraction. When the degree of nitridation was not sufficient, molten Al could not fill the voids in the powder bed, leading to the formation of low-quality composites with high porosities. However, excessive nitridation rapidly consumed the nitrogen gas, leading to a rapid drop in the pressure of the crucible and revealing the remaining aluminum powder in the upper part of the powder bed. Hence, it is important to optimize the degree of nitridation by manipulating the characteristics of the starting materials (e.g., chemical composition of the aluminum powder and type, morphology, and volume fraction of the ceramic reinforcement) and the processing variables (e.g., processing temperature and time, and the flow rate of nitrogen gas). A systematic study to examine the combined effect of each variable on the nitridation behavior is required to determine the optimal conditions for producing AMCs with desirable properties for target applications.

## Figures and Tables

**Figure 1 materials-13-01309-f001:**
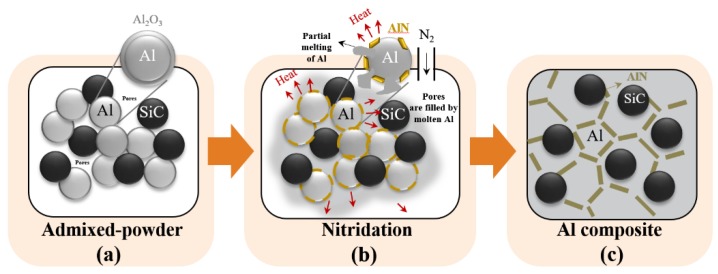
A schematic diagram of the nitridation-induced self-formed aluminum composite (NISFAC) process: (**a**) admixed powder, (**b**) nitridation, and (**c**) Al composite.

**Figure 2 materials-13-01309-f002:**
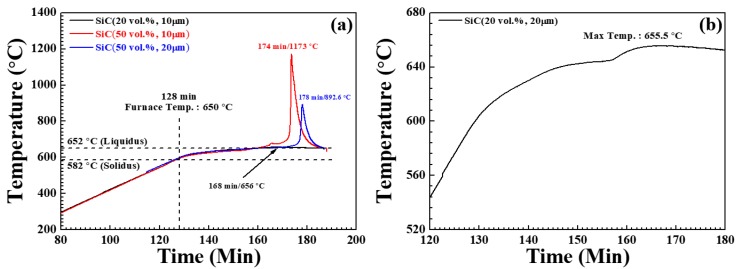
Temperature variation of SiC/6061Al composites during heating at 650 °C in nitrogen gas: (**a**) various volume fractions and sizes of SiC (20 and 50 vol.%, 10 and 20 μm); (**b**) 20 vol.%, 20 μm SiC.

**Figure 3 materials-13-01309-f003:**
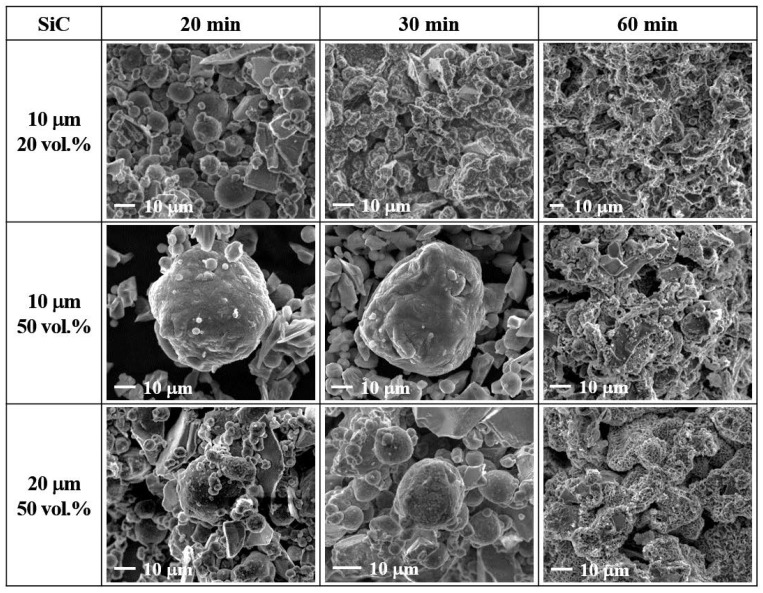
SEM images of SiC/6061Al composites obtained by heating at 650 °C for 20, 30, and 60 min.

**Figure 4 materials-13-01309-f004:**
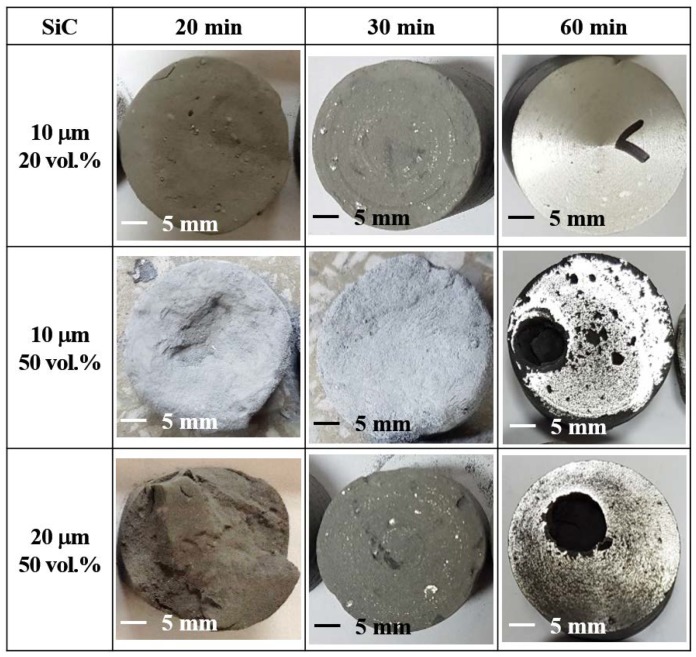
Photographic images of SiC/6061Al composites obtained by heating at 650 °C for 20, 30, and 60 min after lathe working.

**Figure 5 materials-13-01309-f005:**
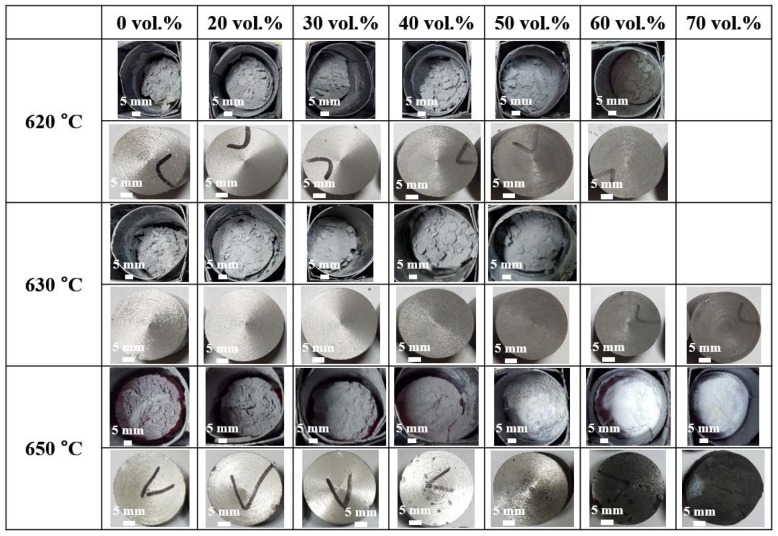
Photographic images of the 0–70 vol.% SiC (20μm)/6061Al composites in their crucibles immediately after heating for 1 h at various temperatures (620–650 °C) and air-cooling (upper rows), and images of the composites after machining (lower rows).

**Figure 6 materials-13-01309-f006:**
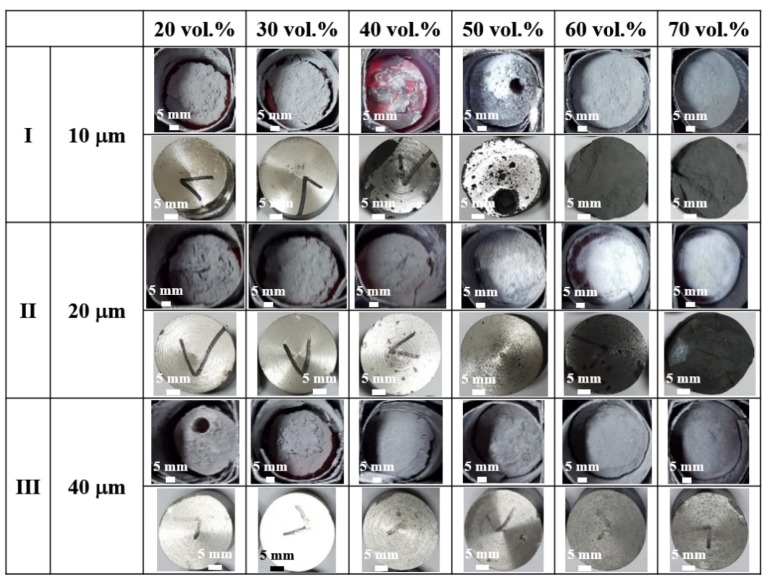
Photographic images of the 20–70 vol% SiC (10–40 μm)/6061Al composites in their crucibles immediately after heating for 1 h at 650 °C and air-cooling (upper rows), and photographic images of the composites after machining (lower rows).

**Figure 7 materials-13-01309-f007:**
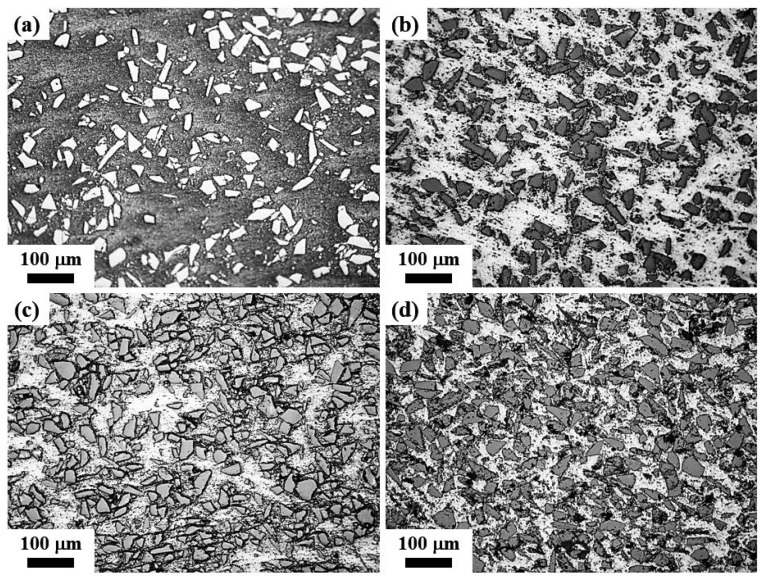
Optical microscope images of the SiC/6061Al composites produced by the NISFAC process in quantities of 500 g (**a**,**b**) and 1000 g (**c**,**d**).

**Table 1 materials-13-01309-t001:** Degree of nitridation and temperature of the SiC/6061 Al composite powder bed after heating at 650 °C for 20, 30, and 60 min.

SiC	20 min		30 min		60 min		Max Temp. (°C) /Time (min)
	Degree of Nitridation	Temperature of Bed (°C)	Degree of Nitridation	Temperature of Bed (°C)	Degree of Nitridation	Temperature of Bed (°C)	
20 vol.% 10 μm	2.3	640	4.0	645	5.4	650	656/40
50 vol.% 10 μm	2.1	633	3.8	646	49.8	650	1173/46
50 vol.% 20 μm	2.7	638	4.2	648	31.3	650	893/50

**Table 2 materials-13-01309-t002:** Degree of nitridation of the 0–70 vol% SiC (20 μm)/6061Al composites heated at 620–650 °C for 1 h.

	0 vol.%	20 vol.%	30 vol.%	40 vol.%	50 vol.%	60 vol.%	70 vol.%
620 °C	2.3	3.8	4.3	7.1	10.2	9.4	
630 °C	2.1	3.3	5.0	5.6	6.2	8.3	12.3
650 °C	4.6	4.4	5.2	11.5	33.6	66.7	94.2

**Table 3 materials-13-01309-t003:** Degree of nitridation of the 0–70 vol% SiC (10–40 μm)/6061Al composites heated at 650 °C for 1 h.

	20 vol.%	30 vol.%	40 vol.%	50 vol.%	60 vol.%	70 vol.%
10 μm	5.7	6.9	54.0	49.8	7.7	12.1
20 μm	4.4	5.2	11.5	33.6	66.7	94.2
40 μm	4.7	5.6	6.3	7.7	9.0	16.9

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
