# Peer review of "Effect of Material and Process Variables on Characteristics of Nitridation-Induced Self-Formed Aluminum Matrix Composites—Part 1: Effect of Reinforcement Volume Fraction, Size, and Processing Temperatures"

_materials, 2020, doi:10.3390/ma13061309_

Round 1

Reviewer 1 Report

1. Title - no significant effects and dependencies have been demonstrated.
2. Abstract is more like a brief description of a patent. The scientific essence of the work is not revealed.
3. Figure 3 and figure 5 are not informative. SEM images are very small. A large number of images of samples without a microstructure is not of scientific interest.
4. EDS is not enough to represent the chemical composition of the samples. It is unclear in what crystalline state silicon carbide and aluminum nitride are. In addition, the EDS is not accurate enough for quantitative phase analysis.
5. Interesting scientific results were presented by the authors in [16-21], which is not in this paper. There is no confirmation or refutation of the mechanisms described by the authors in previous works using similar research methods.
6. The relevance is doubtful, since the authors cite only their own works as modern (last 5 years) articles in this field.

Reviewer 2 Report

Article deals with in-situ synthesis of aluminum metal matrix composites. Al and SiC powders are mixed and melted in nitrogen atmosphere.

Several points authors need to clarify:

Al powder is oxidized and after melting quality is low because of this reason. Dissolved gas concentration in solid aluminum is much lower than in liquid thus remaining gases is problem in aluminum industry. 7 micron Al powder heavily oxidizes if exposed to air. Authors use volume fractions for Al/SiC powder quantification. For fine powder this is not objective measure. I think it would be better to use weight %. 3 contains too many small pictures, it would be better to use fewer pictures, but better explain what is shown. 5. I think this picture is not necessary. Why show 20 similar metal cylinders and powder containers in scientific article? Main result of the article is summarized in Fig. 6. High SiC concentration in al is visible. Typical size of the particle is 30-50 microns. In my opinion this result needs to be better explained: Predicted quality of this material, possibilities of scaling, economic feasibility (in the beginning it is said that this could be used for high-tech and for everyday life as well). Of course, hardness, tensile strength and other mechanical tests would be great to show the performance of the material. In this article authors aim for very high particle concentration. Why ? In many other works it is up to few percent and it is reported that even too high can decrease material properties. Can this method be used to for TiB and TiN particle mixing in aluminum ?

Reviewer 3 Report

The paper is quite readable. 

Since several parameters are studied, I would suggest to use a formal DOE method and to use quantitative outputs from the images.

Regarding the figures themselves, figure n. 4 is missing in the manuscript and, since it is a key-point to understand the results, in any case a resubmission is needed.

Round 2

Reviewer 1 Report

Authors' answers and corrections allow a positive revision of the manuscript

Author Response

Thank you for your review.

Reviewer 3 Report

Now the paper can be published.

Author Response

Thank you for your review.